# Effect of Cdx2 Polymorphism on the Relationship between Dietary Calcium Intake and Peak Bone Mass in Young Japanese Women

**DOI:** 10.3390/nu12010191

**Published:** 2020-01-10

**Authors:** Fumi Oono, Yuri Sakamoto, Yoichi Tachi, Hideaki Mabashi-Asazuma, Kaoruko Iida

**Affiliations:** 1Department of Nutrition and Food Science, Graduate School of Humanities and Sciences, Ochanomizu University, Tokyo 112-8610, Japan; g1940518@edu.cc.ocha.ac.jp (F.O.); mabashi.hideaki@ocha.ac.jp (H.M.-A.); 2Laboratory of Nutrition Physiology, Tokyo Kasei University, Tokyo 173-8602, Japan; sakamoto-y@tokyo-kasei.ac.jp (Y.S.); tachi@tokyo-kasei.ac.jp (Y.T.); 3Institute for Human Life Innovation, Ochanomizu University, Tokyo 112-8610, Japan

**Keywords:** vitamin D receptor gene, Cdx2 polymorphism, bone mass, premenopausal women, calcium consumption

## Abstract

Studies investigating the effect of the caudal-type homeobox protein 2 (Cdx2) polymorphism in the vitamin D receptor gene and calcium intake on bone mass have shown inconsistent results. This study investigated whether the effect of calcium intake on peak bone mass is affected by Cdx2 polymorphism in young Japanese women. A cross-sectional study of 500 young women was conducted. Dietary intake was assessed by the Food Frequency Questionnaire. The osteo sono-assessment index (OSI), assessed by the qualitative ultrasound method, was used as a bone mass index. The subjects were divided into two groups by the median calcium intake. The OSI was not different among Cdx2 genotypes and between calcium groups (*p* = 0.960, *p* = 0.191, respectively). The interaction between calcium and Cdx2 genotypes on the OSI approached significance (GG versus GA and AA genotypes, *p* = 0.092). The difference in the OSI between calcium groups was significant in the GG genotype (*p* = 0.028), but not in the GA or AA genotypes (*p* = 0.501, *p* = 0.306, respectively). Adjustment for covariates (body mass index and physical activity) did not change the results. In conclusion, the relationship between dietary calcium intake and peak bone mass may vary according to Cdx2 polymorphism.

## 1. Introduction

Osteoporosis is becoming a major public health problem worldwide, particularly in the aging community. Women, specifically, have a high risk of developing osteoporosis because of a lower peak bone mass than men [1] and a sharp decrease in bone mass after menopause [2]. In addition to preventing bone loss during aging, increasing the peak bone mass at a younger age is also important for the prevention of osteoporosis and reducing the risk of fracture [3].

Determinants of peak bone mass include lifestyle factors, such as physical activity [4,5] and calcium intake [4,6], as well as genetics [6,7]. Calcium intake is widely accepted as an important nutritional factor for increasing peak bone mass, and the national osteoporosis foundation regards calcium intake as the best grade of evidence for increasing peak bone mass [4]. However, the relationship between dietary calcium intake and bone mass (or density) is obscure [8,9,10]. The potential interactions between calcium intake and genetics, such as the vitamin D receptor (VDR) polymorphism on bone mass, have been reported during the last few decades [11,12], and Eisman [11] suggested the necessity of personalizing nutritional recommendations according to each genotype.

The caudal-type homeobox protein 2 (Cdx2) polymorphism would be important among VDR polymorphisms because it is located in the promoter region of the VDR gene and a previous study reported that A allele relates to higher promoter activity in vitro [13]. However, the results of studies focusing on the interaction of the Cdx2 polymorphism and calcium intake on bone mass (or density) are inconsistent [14,15,16,17]. The inconsistency can be partly explained by the difference in study populations. Genotype frequencies, linked genes, environmental factors, and dietary factors are different among studies. Uitterlinden et al. [18] suggested that different populations can have different gene-environment interactions. With regard to premenopausal women, only two studies have investigated the effect of the Cdx2 polymorphism and calcium intake on bone mass [15,17]; one focused on girls in the adolescence stage before reaching peak bone mass (12.3–17.9 years old) [15], and the other on women over a wide range of age (20–54 years old) [17]. This age heterogeneity might lead to unclear results because menarcheal status, experience in delivery, and the effects of other lifestyle factors (e.g., smoking, alcohol intake, and physical activity) would vary in such a diverse population.

We, therefore, conducted a cross-sectional study among young women, whose bone mass was considered to be stable, to investigate the effect of Cdx2 genotypes on the relationship between dietary calcium intake and peak bone mass.

## 2. Materials and Methods

### 2.1. Study Population

The subjects were 555 female college and junior college students aged 20–24 years old, who were recruited between 2015 and 2019. Written informed consent was obtained from all subjects. The study was conducted in accordance with the Declaration of Helsinki, and the study was approved by the ethics committee of Tokyo Kasei University (approval number: ITAH26-04).

We excluded the following subjects from analysis: those who received obstetrics treatment such as estrogen or steroid hormone therapy (*n* = 42), those who had missing data (*n* = 9), and those who reported a daily energy intake less than half of their required energy intake (*n* = 4) or more than 1.5 times their required energy intake (*n* = 0). Required energy intake was determined according to the Japanese Dietary Reference Intake 2015 [19] using the medium physical activity category and the individual age, height, and weight. Overall, 500 subjects were included in the analysis.

### 2.2. Dietary Assessment

Dietary intake was assessed using a Food Frequency Questionnaire based on food groups (FFQg) Ver.3.0, which consists of questions about 29 food groups and 10 types of cooking to estimate energy and nutrient intake during the past 1–2 months [20,21]. Excel add-in software (Excel Eiyou-kun, Kenpakusha Co. Ltd., Tokyo, Japan) was used to analyze the data. Daily energy and calcium intake estimated by the FFQg significantly correlated to that estimated by 7-days weighed dietary records (*r* = 0.465, 0.410, respectively) [20,21].

### 2.3. Genotyping

Genomic DNA was extracted from saliva using an Orange^®^ DNA extraction kit (DNA Genotek Co Inc., Ottawa, OT, Canada) according to the manufacturer’s protocol. Extracted DNA was diluted to a concentration of less than 100 ng/µL and stored at −20 °C.

Genotypes were detected by real-time polymerase chain reaction (PCR) using the Thermal Cycler Dice^®^ Real Time System (TaKaRa Bio Inc, Shiga, Japan). The Cdx2 polymorphism (rs11568820) was assessed using the following primers: 5′-AAGGAAAGAAAGAAAGGAAG-3′ (forward) and 5′-GGTCTTCCCAGGACAGTAT-3′ (reverse), and the probes: FAM-(Eclipse)AGGTCACArGTA and ROX-(Eclipse)GGTCACArATAA. The reaction mixture contained 12.5 µL Cycleave PCR Reaction Mix (2×) (TaKaRa Bio, Shiga, Japan), 0.5 µL of each primer (10 µM), 0.1 µL of each probe (50 µM), 1.0 µL sample DNA, and 11.3 µL water. PCR was performed under the following conditions: 95 °C for 10 s and then 50 cycles of amplification (95 °C for 5 s, 55 °C for 10 s, and 72 °C for 20 s).

### 2.4. Measurement of Calcaneal Bone Mass

Calcaneus bone mass was assessed using the quantitative ultrasound (QUS) method on the right calcaneus using an AOS-100SA system (Hitachi Co. Ltd., Tokyo, Japan). The previous reports showed that QUS measurements are significantly correlated with bone mineral density assessed by dual-energy X-ray absorptiometry (DXA, *r* = 0.804, *p* < 0.001) [22], and are able to predict the risk of fracture [23,24].

We used the osteo sono-assessment index (OSI) as an index of calcaneus bone mass. The OSI was obtained from the transmission index (TI) and the speed of sound (SOS) using the following formula: OSI = TI × SOS^2^. OSI is highly reproducible [25]. All OSI values presented in this paper were divided by 10^6^.

### 2.5. Assessment of Other Variables

As a previous study reported that physical activity in junior high school strongly affects peak bone mass [26], participants were required to complete a self-administered questionnaire regarding their physical activity habits in junior high school. Physical activity habits were defined as exercise habits other than physical education class and subjects were asked what type of sports they undertook in each period. Height and weight were also obtained by self-administered questionnaire, and then, body mass index (BMI) (kg/m^2^) was calculated by dividing the weight by the square of the height.

### 2.6. Statistical Analysis

All statistical analyses were performed using SPSS version 24 software (SPSS Inc, Chicago, IL, USA). Accordance with Hardy-Weinberg’s equilibrium was assessed using χ^2^ tests. Calcium intake was adjusted for daily energy intake using the density method. The value obtained from the density method was converted to an integer. The subjects were divided into two groups by the median calcium intake.

To determine the difference in characteristics based on genotypes and calcium intake groups, analysis of variance (ANOVA) and independent-samples *t*-test were employed, respectively. χ^2^ tests were used for categorical variables (i.e., physical activity habits). Analysis of covariance (ANCOVA) was used to examine differences in the OSI among Cdx2 genotypes or between calcium intake groups adjusted by covariates; the covariates were BMI (kg/m^2^, continuous) and physical activity habits in junior high school (yes or no).

The relationship between genotypes and calcium intake on the OSI was evaluated by two-way ANOVA and adjusted with the above-mentioned covariates. To assess the effect of calcium intake on OSI based on Cdx2 genotypes, we separated subjects according to Cdx2 genotypes and used ANCOVA. All reported *p* values were two-tailed, and *p* < 0.05 was considered significant.

## 3. Results

The basic characteristics of the subjects are shown in Table 1. All subjects were female and the mean value of the OSI was 2.72.

The genotype frequency (Table 2) was in the Hardy-Weinberg equilibrium (*p* = 0.163). There were no significant differences in all variables among Cdx2 genotypes. The OSI among the three genotype groups showed no significant difference when adjusted with the covariates (BMI and physical activity habits in junior high school) (*p* = 0.648).

As the median calcium intake among our subjects was 250 mg/1000 kcal, the subjects were assigned to either the low group (≦250 mg/1000 kcal) or the high group (≧251 mg/1000 kcal). The characteristics of each group are shown in Table 3. There was no significant difference in all variables between calcium groups. The OSI between calcium groups showed no significant difference even after adjustment with the covariates (*p* = 0.320).

The interaction between calcium intake and Cdx2 genotype showed a tendency of significance (*p* = 0.092 and *p* = 0.079, not adjusted and adjusted, respectively) (GG versus carriers of mutant type A gene). The difference in the OSI between calcium intake groups was significant for the GG genotype (*p* = 0.028), but not for the GA or AA genotypes (*p* = 0.501, *p* = 0.306, respectively). Adjustment for covariates did not change the results (Table 4).

## 4. Discussion

In young females, after achieving peak bone mass of the calcaneus [27], we found that the relationship between calcium intake and peak bone mass differed among Cdx2 genotype groups.

The Cdx2 polymorphism did not affect bone mass in this study. The genotype frequency in this study was in accordance with those of the Hardy-Weinberg equilibrium and previous studies involving Japanese participants [13,17]. While many studies have not demonstrated a relationship between the Cdx2 polymorphism and bone mineral density [13,14,16,17,28,29,30,31,32,33,34,35], some have identified relationships [36,37]. This inconsistency might be due to relatively small study sample sizes or a wide range of age. When we focused on the studies only among premenopausal females, they showed that bone mineral density did not differ according to Cdx2 genotype [13,17,29].

Mean OSI values did not differ between dietary calcium intake groups. While previous studies have reported a significant relationship between calcium intake and bone mineral density at several bone sites in young females [38,39,40,41,42], some of them found no significant relationship at other bone sites [40,41,42]. Moreover, Correa-Rodríguez M et al. [43] suggested that the calcaneus was more affected by physical activity than nutrient intake; here, the OSI was affected by the physical activity habits in junior high school (data not shown). Therefore, a possible explanation for the result of no relationship between calcium intake and the OSI is that a very weak (or null) effect of calcium intake on calcaneal peak bone mass was masked by physical activity and other covariates.

However, upon division of the subjects according to Cdx2 polymorphism genotype, the effect of calcium intake on bone mass became obvious. We found that subjects with higher calcium intake had approximately 3% higher peak bone mass than those with lower calcium intake, only in subjects with the GG genotype. Hernandez et al. [44] suggested that a 10% increase in peak bone mass is predicted to delay the onset of osteoporosis by more than ten years. Therefore, the increase in GG subjects would be expected to delay the onset of osteoporosis by a few years.

Consistent with our results, Morita et al. [17] reported that higher milk intake relates to higher bone mineral density only in the GG genotype among premenopausal Japanese women. However, a study among European children showed that there was no interaction between calcium intake and Cdx2 polymorphism on bone mineral density [15]. This disparity is partly because of the small number (*n* = 117) and younger age (mean age: 14.9 years old) of subjects in the study [15].

Mechanistically, Arai et al. [13] showed that the A allele of Cdx2 has a higher transcriptional activity of VDR than the G allele in vitro. To our knowledge, however, there is no experimental study investigating whether the Cdx2 polymorphism affects calcium absorption, despite it having been investigated for other VDR polymorphisms (*BsmI*, *TaqI*, and *ApaI*) using dual stable isotopes [45,46]. It was reported that the Cdx2 polymorphism modified the positive association between calcium intake and prostate cancer risk in White [47] and African American [48] people. Therefore, there is a possibility that the Cdx2 polymorphism changes calcium metabolism according to the amount of calcium intake.

One of the advantages of this study was that the age range of subjects was very narrow (20 ± 0.3 years old) and 97% of subjects (*n* = 486) were 20 years old. Therefore, we were able to reduce the effect of factors affecting peak bone mass (e.g., alcohol intake, smoking, and reproductive history) because alcohol intake and smoking by people under 20 years old are prohibited by law in Japan. Moreover, a previous study showed that alcohol intake, smoking, and reproductive history do not affect bone mineral density among young women aged 19–25 years old [49].

This study has several limitations. First, statistical power to detect the difference in OSI values between calcium groups for each Cdx2 genotype is different because the number of subjects in each genotype group is not the same. Therefore, the relationship between calcium intake and OSI in each genotype should be interpreted with caution.

Second, we did not measure serum vitamin D concentrations, which is considered an important factor for bone health. Although meta-analysis suggested the relationship between serum vitamin D and fracture risk, the relationship between serum vitamin D concentrations and bone status in young Japanese women has not reached a consensus [40,50]. Moreover, some studies have reported no significant difference in serum vitamin D concentration between Cdx2 genotypes [16,29].

Third, the validation of calcium intake assessed by FFQg was reported only for crude values (mg/day) and not according to the density method (mg/1000 kcal) [21]. Moreover, the validity was confirmed by the correlation coefficient [21], and the validity of the absolute quantitative value remained unclear. Therefore, we used calcium intake to categorize participants to reduce the effect of measurement error, while categorized calcium intake was used as a categorical independent variable for ANCOVA.

Fourth, some residual confounding may exist. BMI was calculated from self-reported height and weight. The previous study suggested systematic bias in self-reporting height and weight values [51]. Therefore, some subjects, especially those with obesity, may underreport their weight and then the respective BMI values might be inaccurate. Moreover, the questionnaire regarding physical activity habits was not validated. Although the proportion of subjects who referred physical activity habits almost match the results of a survey made by the Japan Sports Agency published in 2016 [52], we cannot entirely rule out the possibility of misclassification.

Finally, we assessed only the calcaneus using the QUS method as an index of peak bone mass. Although a previous study reported the values assessed by the QUS method significantly correlate to those assessed by DXA on other bone sites [53], future studies using additional bone sites are required to ascertain whether the genotype difference in the effect of calcium intake on peak bone mass exists elsewhere.

## 5. Conclusions

This study, conducted in a narrow-aged young Japanese women cohort, suggests that the relationship between dietary calcium intake and peak bone mass is affected by Cdx2 genotypes. A homogeneous population can be an effective way to reduce confounding and to detect functional relationships. However, generalizability to the entire population is limited. Further genetics-based studies, focused on the effects of nutrients on peak bone mass, in various homogeneous populations are required to increase peak bone mass more effectively.

## Figures and Tables

**Table 1 nutrients-12-00191-t001:** Characteristics of participants in this study (*n* = 500).

Variable	Value
Age (years)	20.0 ± 0.3
Height (cm)	158.7 ± 5.3
Weight (kg)	51.4 ± 6.0
BMI (kg/m^2^)	20.4 ± 2.0
OSI	2.72 ± 0.26
Energy (kcal/day)	1723 ± 388
Calcium (mg/day)	456 ± 169
Calcium (mg/1000 kcal)	263 ± 69
Physical activity habits in junior high school (%)	66.6

Values are means ± standard deviation or percentage. Abbreviations: BMI, body mass index; OSI, osteo sono-assessment index.

**Table 2 nutrients-12-00191-t002:** Characteristics of participants compared to caudal-type homeobox protein 2 (Cdx2) genotyping.

	Cdx2			*p*
	GG	GA	AA	
Number (%)	159 (31.8)	264 (52.8)	77 (15.7)	-
Age (years)	20.0 ± 0.1	20.1 ± 0.4	20.0 ± 0.2	0.125
Height (cm)	158.3 ± 5.7	159.0 ± 5.1	158.2 ± 5.2	0.242
Weight (kg)	51.2 ± 6.1	51.4 ± 5.9	52.0 ± 6.1	0.669
BMI (kg/m^2^)	20.4 ± 2.0	20.3 ± 2.1	20.7 ± 2.0	0.248
OSI	2.72 ± 0.25	2.71 ± 0.27	2.71 ± 0.26	0.960
Calcium (mg/1000 kcal)	259 ± 66	266 ± 72	261 ± 64	0.579
Physical activity habits in junior high school (%)	62.9	67.4	71.4	0.393

Values are means ± standard deviation or percentage. The *p* values are the level of significance of difference based on genotypes. The *p* values of continuous variables were assessed by analysis of variance (ANOVA) and the *p* values of physical activity habits were assessed by the χ^2^ test. Abbreviations are the same as in Table 1.

**Table 3 nutrients-12-00191-t003:** Characteristics of participants compared to dietary calcium intake (mg/1000 kcal).

	Dietary Calcium Intake		*p*
	Low (*n* = 248)	High (*n* = 252)	
Calcium (mg/1000 kcal)	211 ± 28	314 ± 58	-
Age (years)	20.0 ± 0.2	20.1 ± 0.4	0.103
Height (cm)	158.5 ± 5.6	158.8 ± 5.1	0.546
Weight (kg)	51.2 ± 6.2	51.7 ± 5.8	0.351
BMI (kg/m^2^)	20.4 ± 2.1	20.5 ± 1.9	0.500
OSI	2.70 ± 0.27	2.73 ± 0.25	0.191
Energy (kcal/day)	1695 ± 384	1752 ± 390	0.101
Physical activity habits in junior high school (%)	64.9	68.3	0.449

Values are means ± standard deviation. The *p* values are the level of significance of difference based on dietary calcium intake. The *p* values of continuous variables were assessed by an independent-samples *t*-test and the *p* values of physical activity habits were assessed by the χ^2^ test. Abbreviations are the same as in Table 1.

**Table 4 nutrients-12-00191-t004:** Difference in OSI between calcium intake groups based on Cdx2 genotypes.

Cdx2 Genotypes	Calcium Intake	*n*	Not Adjusted		Adjusted ^†^	
			OSI	*p*	OSI	*p*
GG	Low	81	2.68 ± 0.03	0.028 *	2.68 ± 0.03	0.047 *
	High	78	2.76 ± 0.03		2.76 ± 0.03	
GA	Low	127	2.70 ± 0.02	0.501	2.71 ± 0.02	0.760
	High	137	2.72 ± 0.02		2.72 ± 0.02	
AA	Low	40	2.74 ± 0.04	0.306	2.74 ± 0.04	0.248
	High	37	2.68 ± 0.04		2.68 ± 0.04	

Values are means ± standard error. The *p* values are the level of significance of difference in the OSI between dietary calcium intake groups. Analysis was performed using analysis of covariance (ANCOVA). ^†^ adjusted for BMI and physical activity habits in junior high school. * *p* < 0.05.

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
