# Peer review of "Effect of Cdx2 Polymorphism on the Relationship between Dietary Calcium Intake and Peak Bone Mass in Young Japanese Women"

_nutrients, 2020, doi:10.3390/nu12010191_

Round 1

Reviewer 1 Report

The authors have submitted a well-written and interesting report on an important topic area. The research benefits from an good sized sample in a population where research into bone health is of paramount importance. Please see specific comments below:

Introduction, Lines 64-65: The statement in this sentence "among young women who's peak bone mass had been achieved" is rather broad as we cannot know whether peak bone was reached for all these women. I would suggest revising the wording of this statement.

2.2 Dietary Assessment: Spelling error on first word ('dietatary' rather than dietary)

2.4 Measurement of calcaneal bone mass, Lines 101-103: Reword this sentence, for e.g." Past research has shown that QUS 101 measurements are significantly correlated to bone mineral density assessed by dual-energy X-ray 102 absorptiometry (DXA, r = 0.804, p < 0.001) [22], and are able to predict the risk of fracture [23,24]"

2.5: Assessment of other variables: Some extra information on the junior high school activity assessment would be useful i.e. Was this a validated survey?

2.5: Assessment of other variables: Was Height and Weight also measured or only asked in the self administered questionnaire? If only asked, this does open some potential for bias and should be mentioned as a limitation in the discussion.

5. Conclusions: The final sentence is a bit vague here - it would be useful if the authors provided a little more info on future directions/next steps in this important area of research.

Reviewer 2 Report

This study by Oono et al. examined the effect of a polymorphism in the promoter region of the Cdx2 gene on bone mass in young Japanese women. Overall the manuscript was well written. Although the authors do mention the results of in vitro studies on the promoter in the discussion section, it might be helpful to the reader to mention the results of these studies in the Introduction. The manuscript would also benefit from a brief discussion in which the authors justify the appropriateness of the statistical methods used.
